# Antimicrobial and Antiherpetic Properties of Nanoencapsulated *Hypericum perforatum* Extract

**DOI:** 10.3390/ph18030366

**Published:** 2025-03-04

**Authors:** Yoana Sotirova, Nadezhda Ivanova, Neli Ermenlieva, Neli Vilhelmova-Ilieva, Lora Simeonova, Miroslav Metodiev, Viliana Gugleva, Velichka Andonova

**Affiliations:** 1Department of Pharmaceutical Technologies, Faculty of Pharmacy, Medical University of Varna, 9000 Varna, Bulgaria; yoana.sotirova@mu-varna.bg (Y.S.); viliana.gugleva@mu-varna.bg (V.G.); velichka.andonova@mu-varna.bg (V.A.); 2Department of Microbiology and Virology, Faculty of Medicine, Medical University of Varna, 9000 Varna, Bulgaria; neli.ermenlieva@mu-varna.bg; 3Department of Virology, Stephan Angeloff Institute of Microbiology, Bulgarian Academy of Sciences, 26 G. Bonchev Str., 1113 Sofia, Bulgaria; nelivili@gmail.com (N.V.-I.); losimeonova@gmail.com (L.S.); mnmetodiev@microbio.bas.bg (M.M.)

**Keywords:** antibacterial activity, antifungal activity, antimicrobial activity, antiviral activity, herpesvirus, hyperforin, lipid nanoparticles, nanostructured lipid carriers, phloroglucinols, St. John’s wort

## Abstract

**Background/Objectives:** This study aims to gain insights into the antimicrobial and antiherpetic activity of hyperforin-rich *Hypericum perforatum* L. (HP) extract using nanostructured lipid carriers (NLCs) as delivery platforms. **Methods**: Two established NLC specimens, comprising glyceryl behenate and almond oil or borage oil, and their extract-loaded counterparts (HP-NLCs) were utilized. Their minimal bactericidal/fungicidal concentrations (MBC; MFC) were investigated against *Escherichia coli* ATCC 25922, *Staphylococcus aureus* ATCC 25923, *Pseudomonas aeruginosa* ATCC 10145, *Klebsiella pneumoniae* ATCC 10031, and *Candida albicans* ATCC 10231. The anti-herpesvirus (HSV-1) potential was evaluated concerning antiviral and virucidal activity and impact on viral adsorption. **Results**: The borage oil-based extract-loaded nanodispersion (HP-NLC2) exhibited pronounced microbicidal activity against *S. aureus* (MBC 6.3 mg/mL), *K. pneumoniae* (MBC 97.7 µg/mL), and *C. albicans* (MFC < 48.8 µg/mL), unlike the almond oil-containing sample (HP-NLC1), which showed only weak inhibition of the fungal growth. HP-NLC2 was found to be less cytotoxic and to suppress HSV-1 replication slightly more than HP-NLC1, but generally, the effects were weak. Neither the empty lipid nanoparticles nor the HP extract-loaded carriers expressed activity against *E. coli*, *P. aeruginosa*, the HSV-1 extracellular virions, or viral adhesion. **Conclusions**: It could be concluded that both HP-NLC samples revealed only minor antiherpetic potential of the hyperforin-rich extract, but HP-NLC2 demonstrated significant antibacterial and antimycotic activity. Therefore, the latter was featured as a more convenient HP-carrier system for nano-designed dermal pharmaceutical formulations. Such a thorough investigation of hyperforin-determined anti-HSV-1 effects and antibacterial and antimycotic properties, being the first of its kind, contributes to the fundamental knowledge of HP and reveals new perspectives for the utilization, limitations, and therapeutic designation of its non-polar components.

## 1. Introduction

St. John’s wort (*Hypericum perforatum* L.; HP) has a history as a medicinal plant dating back to ancient times. Apart from its well-known application in mental disorders (depression, melancholy, anxiety, and hyperarousal), it has also been used for treating various skin pathologies such as hemorrhages, infections, inflammations, burns, wounds, bruises, etc. [1,2,3]. The technological evolution in scientific research satisfies the growing interest in the herb; multiple biochemical, pharmacological, pharmacokinetic, and clinical trials have been conducted and have identified its antioxidant, anti-inflammatory, antibacterial, and antiviral properties, as well as the mechanisms underlying them [4,5,6,7]. Moreover, after a detailed phytochemical analysis, it has been revealed that several groups of biologically active compounds (BACs) in St. John’s wort contribute to these effects, viz., phloroglucinols, naphthodianthrones, phenolic acids and depsides, flavonoids, and xanthones [8,9].

Concerning the antibacterial activity of HP, phloroglucinols and naphthodianthrones stand out as the main microbicidal constituents (Figure 1) [5]. Numerous studies testify to the pronounced effect of the herb against Gram-positive bacteria. Among the most susceptible representatives are reported to be *Staphylococcus aureus* (including methicillin-resistant strains), *Bacillus cereus*, and *Enterococcus faecalis* [10,11,12]. Most of the research indicates an absence of activity of St. John’s wort preparations against Gram-negative bacteria and fungi [13,14,15,16]. However, some reports contradict the latter statement by reporting antimicrobial effects against *Escherichia coli*, *Klebsiella pneumoniae*, *Pseudomonas aeruginosa*, and *Candida albicans* [12,17,18].

As for the antiviral properties of St. John’s wort, the herbal substance is reported to be effective against encapsulated viruses—HIV-1, Influenza Virus Type A, Avian Infectious Bronchitis Virus, SARS-CoV-2, as well as others [7,19,20,21]. Interestingly, only naphthodianthrones (hypericin, pseudohypericin) have been tested and proven to exhibit anti-herpesvirus activity. Their action is determined to a significant extent by photoactivation upon daylight exposure [22]. However, when activated in the presence of oxygen, naphthodianthrones become powerful generators of singlet oxygen, superoxide anions, and other reactive oxygen species. The latter activate various cytotoxic cascades in cells, the ultimate result of which is cell death. This effect is used in the so-called photodynamic therapy but may turn into a “defect” in other skin pathological conditions [23].

A major hindrance to the topical application of HP phloroglucinols is their inherent chemical instability. In particular, the main phytochemical from that class, hyperforin, is photosensitive and highly reactive to oxygen [24]. A successful approach for preserving such labile BACs is their incorporation into nanoscale drug-delivery systems. Among the first such created were polymer and metal nanoparticles, liposomes, and nanoemulsions [25]. In the early 1990s, the Müller and Gasco teams pioneered the idea of solid lipid nanoparticles (SLNs) [26,27]. SLNs are presented as a safer, non-toxic alternative to other types of nanoparticles. Moreover, they are physically more stable than liposomes and, in addition, their production is more cost-effective [28]. SLNs have a solid lipid core surrounded by a stabilizing surfactant layer. Their key advantages as drug carriers are biocompatibility, biodegradability, lack of toxicity, and the possibility of modified drug release [29]. Their main disadvantages, namely low drug-loading capacity and rapid drug release, are related to their highly organized crystalline lipid matrix [30]. To overcome the aforementioned drawbacks of SLNs, the so-called nanostructured lipid carriers (NLCs) were subsequently introduced [31]. In contrast to SLNs, their stability is increased, and their drug-loading capacity is much higher as well. The NLCs’ superiority over SLNs is determined by the liquid lipids included in the cores of the former [30].

Our previous work aimed to obtain a hyperforin-rich St. John’s wort extract and incorporate it into a suitable NLC system prior to topical application in wound healing [32,33]. To the best of our knowledge, our study on isolating and “preserving” such an extract by a lipid nanocarrier-based approach is the first of its kind. The herein presented work on the antimicrobial activity of the nanocarriers in question is also sui generis. We have already proven the wound-healing properties of HP-NLCs. Here, we aim to further emphasize their therapeutic importance by investigating their effect against “classic” pathogens causing wound infections—*S. aureus*, *E. coli*, *K. pneumoniae*, *P. aeruginosa*, and *C. albicans*. We also believe that preserving the main phloroglucinol in the extract via nanoencapsulation would allow us to acquire insights into its anti-herpesvirus potential.

## 2. Results

### 2.1. “Blank” and HP-Loaded NLCs

All data concerning the HP extraction, NLC formulation, and characterization have already been published elsewhere [32,33]. We summarize herein some of the critical findings that could be related to the current results.

The concentration of hyperforin in the HP extracts was found to be 8.9 µg/mL, i.e., its extraction yield was 44.35%; knowing that, the hyperforin content, and not the total HP extract content, was applied for the calculation of the entrapment efficiency (EE) [32]. The HP extract was incorporated into the NLCs in a concentration of 1.25% (*w*/*w*) as according to the Attenuated Total Reflectance (ATR)-FTIR study, higher amounts led to the formation of extract–nanoparticle co-aggregates [32]. The “blank” nanodispersions (NLC1 and NLC2) were observed as milky-white liquids, while the HP-loaded ones (HP-NLC1 and HP-NLC2) inherited the extract’s green shade (Figure 2A). Dynamic Light Scattering (DLS) analysis revealed an initial mean hydrodynamic size of about 150 nm (Figure 2B). The EE (%), derived from the ratio of the theoretically loaded hyperforin to the actual HPLC-detected hyperforin, was in the range of 70.44–74.49% (Figure 2C). Further, Transmission Electron Microscopy (TEM) discovered the particles’ irregular shape and well-defined core and shell. In the presence of the plant extract, the bilayer structure of the nanoparticles appeared to be more amorphous, but the encapsulation did not affect their matrix organization or integrity (Figure 2D). Except for HP-NLC1, the nanocarriers retained their average particle diameter below 200 nm during one month of storage at 4 °C (Figure 2B). The almond oil-based extract-loaded nanoplatforms also demonstrated a significant deterioration of homogeneity, expressed as the highest polydispersity index (Figure 2B). Moreover, their ability to encapsulate and preserve HP extract was also lower than that of HP-NLC2. However, in the search of a better choice for the most suitable HP carrier, despite the obvious advantageous physicochemical properties of the borage-based HP-NLC specimen so far, we decided to subject both samples to antimicrobial and antiviral tests.

### 2.2. Antiviral Activity

Before conducting the antiviral experiments, we established the non-toxic range of the investigated samples to exclude a toxic effect on the cells. NLC1 and NLC2 showed no toxicity even at the highest concentration tested. Nanoparticles carrying the active extract showed cytotoxicity close to that of the reference Acyclovir {[9-(2-hydroxyethoxymethyl)-guanine]; ACV}, but slightly stronger—in the range of 2.5–3 times more toxic. HP-NLC2 demonstrated lower cytotoxicity against Madin–Darbey bovine kidney (MDBK) cells compared with HP-NLC1 (Table 1).

The anti-herpesvirus activity of the studied nanocarriers was determined within the defined non-toxic concentration range. The two “blank” specimens did not show any effect on human herpes simplex virus type 1 (HSV-1) replication. The two samples loaded with the extract demonstrated a weak effect on the viral replication of HSV-1 with selectivity index (SI) ≤ 4. However, a stronger effect was established for HP-NLC2 with SI = 4.0 (Table 1).

“Blank” and loaded NLCs were also examined for their impact on extracellular HSV-1 virions. The effect was monitored for different time intervals within 120 min. In all the time intervals studied, none of the samples induced a significant drop in the viral titer (Table 2), indicating that they act only as carriers that do not release the extract outside the cell.

The results obtained so far show that the effect of HP-NLCs occurs only intracellularly, but we decided to check the possibility that they have an effect on the step of virus adsorption by influencing the cell membrane and the binding of cell receptors to the viral structures responsible for the attachment of the virus to the cell. Time intervals of different duration within the viral adsorption (60 min) were investigated. None of the samples showed an effect on this stage at any of the time intervals examined (Table 3). This is more evidence that the studied nanocarriers exert their effect only intracellularly.

### 2.3. Antimicrobial and Antifungal Activity

During this study, all test tubes acquired the opaque off-white appearance inherent in the NLC/HP-NLC suspensions; therefore, determination of minimal inhibitory concentrations by visual assessment of microbial growth-related turbidity was not possible. As a next step, minimal bactericidal/fungicidal concentrations (MBC; MFC) were recorded after transfer to blood agar and another incubation. The NLCs themselves (or their components within the composition) were not found to possess any activity against the targeted pathogenic microorganisms (Figure 3). The control antimicrobial agent, chlorhexidine diacetate, showed MBCs of 35.3 μg/mL (*S. aureus*), 141.2 μg/mL (*E. coli*), 564.8 μg/mL (*K. pneumoniae*), 1130.0 μg/mL (*C. albicans*), and 2260.0 μg/mL (*P. aeruginosa*).

HP-NLC2 demonstrated a distinct microbicidal effect against *K. pneumoniae* and *C. albicans,* with MBC/MFC of 97.7 μg/mL (dilution 1:128) and <48.8 μg/mL (dilution 1:256), respectively. A weaker but still present activity of the nanosuspension in question was recorded against *S. aureus,* with an MBC of 6.3 mg/mL (dilution 1:2) (Figure 4). No antimicrobial activity was established in the extract-loaded nanoparticles against the Gram-negative *E. coli* and *P. aeruginosa*.

## 3. Discussion

The results so obtained testify that the established antimicrobial and anti-herpesvirus action of HP-NLCs is owing to the hyperforin-rich extract and not to the delivery nanoplatforms employed. However, because of the main compound’s extreme chemical instability, the formulation of the extract into the nano-sized lipid particles could be regarded as a requirement for the activity to be manifested. This observation only highlights the practical importance of the current study and the obtained results. Moreover, it explains the reason why standard hyperforin-rich extract cannot be used as a reference formulation.

An ideal nanocarrier system should provide physical endurance over time, high loading capacity, chemical inertness, drug stability or stabilization, lack of toxicity, biocompatibility, and expedient drug permeation and release [34]. According to our previous and current results, the formulation HP-NLC2 fulfills all these criteria and also contributes to an explicit potentiation of the antibacterial and antimycotic activities against *K. pneumoniae*, *S. aureus*, and *C. albicans*. A reasonable explanation for this phenomenon might be sought in the presence of borage oil as the liquid lipid in the nanocarriers’ composition. Several scientific reports have confirmed its antimicrobial action, as well as that of its major polyunsaturated fatty acids, i.e., linoleic and γ-linolenic acids [35], including against the herein-investigated pathogens [36,37,38]. It is likely that the relative content of borage oil within the lipid nanoparticles was below its minimal inhibitory concentrations, and because of this, we did not observe its own antimicrobial effect when applying the “empty” NLC2 sample. We hypothesize that the oil’s permeation-enhancing properties [39] may also have reinforced the HP extract’s activity by facilitating its intracellular passage through the bacterial walls.

The results of any preferential mechanism of action of the hyperforin-rich extract against Gram-positive or Gram-negative pathogens remain inconclusive. It is evident that along with *C. albicans*, the Gram-negative *K. pneumoniae* stands out as the most susceptible to the extract’s action, while the Gram-positive *S. aureus* is affected at much higher doses. Still, the other Gram-negative representatives in this study, *E. coli* and *P. aeruginosa*, demonstrated resistance to the extract’s action in the applied doses. Low susceptibility to antimicrobial agents is unique to *P. aeruginosa*. Its rich resistome, multiple efflux systems, and ability to form biofilms contribute to its high resistance [40]. In addition, the lipopolysaccharides and porin proteins in its outer membrane are distinct from those of the Enterobacteriaceae family, which also ensures its much lower permeability to exogenous substances [41].

There are numerous extraction solvents and techniques that could be applied for the production of St. John’s wort extracts, and they all result in a highly differentiating qualitative and quantitative content of BACs. Unless the herbal preparations are traditional or standardized, their established effects are hard to relate to any other data from the scientific literature [42]. In this regard, directive 2004/24/EC of the European Parliament and of the Council provides a list of several traditional HP preparations. These include the dry extract obtained with the extraction solvent ethanol 45% (*v*/*v*), which contains hypericin of approximately 0.3–0.7% and hyperforin between 0.3 and 1.6%; the liquid extract obtained with the extraction solvent maize oil in a drug–extractant ratio (DER) of 1:13, with an approximate content of hypericin of 0.0013% and hyperforin of 0.01%; the liquid extract obtained with the extraction solvent other vegetable oils (DER 1:4–20), in which hypericin is at least 0.005%; and finally, the ethanolic 45–50% (*v*/*v*) tinctures (DER 1:5 or 1:10), the ethanolic 50% (*v*/*v*) extract (DER 1:2), the expressed juice from the fresh herb, the comminuted herbal substance for tea infusion, and the powdered substance for solid dosage forms [43]. On the other hand, the European Pharmacopoeia (Ph. Eur.) regards the production of quantified St. John’s wort dry extract from the herbal substance by using ethanol (50–80%, *v*/*v*) or methanol (50–80%, *v*/*v*) with a total content of hypericins, expressed as hypericin, in the range of 0.10–0.30%; flavonoids, expressed as rutin, minimum 6.0%; and hyperforin of maximum 6.0% [44]. The herbal substance is defined as a whole or as fragmented, dried flowering tops of HP with a minimal total content of hypericins, expressed as hypericin 0.08%. As the phloroglucinol derivative hyperforin is lipophilic, it is best retrieved by non-polar extractants such as supercritical CO_2_, ethyl acetate, petroleum ether, *n*-hexane, dichloromethane, and *n*-heptane [45,46,47]. Indeed, the highest production yield of hyperforin is achieved with the aid of supercritical CO_2_ extraction; therefore, this method is now also commercially established [46,48]. Taking into account all the above, we related the results of this study with reports providing analytic proof of hyperforin content as the main BAC in the extract or at least using solvents capable of extracting the phloroglucinol.

Drozdov et al. reported the lack of antibacterial activity of St. John’s wort supercritical extract against bacteria and fungi. Interestingly, the authors isolated the pure hyperforin from the extract and demonstrated its antimicrobial effect against *P. aeruginosa* [49]. Naeem et al. established an equivalent efficacy of *n*-butanol and ethyl acetate St. John’s wort extracts against Gram-positive and Gram-negative bacteria (*Bacillus subtilis*, *Sarcina lutea*, *P. aeruginosa*, and *E. coli*) but did not observe any antibacterial activity of *n*-hexane and dichloromethane-retrieved extracts [50]. Schempp et al., on the other side, proved hyperforin to be active only against Gram-positive bacteria, viz., *S. aureus*, *Streptococcus pyogenes*, *Streptococcus agalactiae*, and *Corynebacterium diphteriae* [51]. Similar results were obtained by Avato et al., who found hyperforin and a hyperforin-containing ethyl acetate St. John’s wort extract to be effective against different strains of *S. aureus*, as well as against *B. cereus*, *Bacillus subtilis*, and *E. faecalis*, but ineffective against Gram-negative bacteria and fungi [52]. Despite the relative controversy in these results, it should be acknowledged that the investigated extracts were of different chemical compositions and were applied in different doses. Most importantly, none of the above reports used a nano-delivery system for the hyperforin-containing herbal preparations, and neither did they follow the stability of the phloroglucinol. As for the antiherpetic activity of HP, only the naphthodianthrones in the herb, viz. the hypericins, have been reported so far to possess such [22]; thus, we came to conclude that the current study presents a first thorough investigation of the hyperforin-rich HP extract’s potency against HSV-1.

## 4. Materials and Methods

### 4.1. Materials

All materials used in the study were of pharmaceutical grade. St. John’s Wort air-dried ground flowers, leaves, and shoots were supplied by Bilec Company, Troyan, Bulgaria; anhydrous methylene chloride ≥ 99.7% and sorbitan monooleate were supplied by Thermo Fisher Scientific, Waltham, MA, USA; double-distilled water was obtained in laboratory conditions (Gesellschaft für Labortechnik GmbH, Burgwedel, Germany); almond oil and borage oil were purchased from Alteya Organics, Stara Zagora, Bulgaria; polyoxyethylene (20) sorbitan monooleate was supplied by Sigma-Aldrich, St. Louis, MO, USA. Glyceryl behenate (Compritol^®^ ATO 888) was a kind gift from Gattefossé, Saint-Priest, France.

### 4.2. Methods

#### 4.2.1. Preparation of Hyperforin-Rich HP Extract

The extraction procedure for the preparation of hyperforin-rich HP extract was recently established [32]. The extract was obtained by applying a maceration method under conditions that align with the high reactivity of hyperforin. The extraction was carried out in amber glass flasks under an argon atmosphere in the condition of darkness. Anhydrous methylene chloride was employed as an extractant, and the plant material-to-solvent ratio was set to be 30:100 (*w*/*v*). The extract obtained after separation of the plant residue, filtration, and removal of the organic solvent was dissolved in anhydrous methanol (to a concentration of 20.0 µg/mL) and subjected to an HPLC-UV analysis. The presence of hyperforin was determined by the retention time reported using a hyperforin standard, and its concentration in the isolated extract was determined by the absolute calibration method [32].

#### 4.2.2. Preparation of “Blank” and HP-Loaded NLCs

The optimal carriers for the thus-obtained extract were identified based on a thorough physicochemical characterization of twenty potential specimens by DLS, Laser Doppler Electrophoresis, EE determination, ATR-FTIR Spectroscopy, X-ray Diffraction Analysis, and TEM. The samples currently labeled as NLC1 and NLC2 were recognized to be a “best fit” to the particles’ designation [32]. Herein, a total of four specimens (two “blank”, viz., NLC1 and NLC2, and the corresponding hyperforin-rich St. John’s wort extract-loaded samples, i.e., HP-NLC1 and HP-NLC2) were obtained as previously described [32]. They all comprised a 10% (*w*/*w*) lipid phase with a glyceryl behenate-to-almond oil (NLC1 and HP-NLC1) or glyceryl behenate-to-borage oil (NLC2 and HP-NLC2) ratio of 7:3. The employed surfactant mixture (5% *w*/*w*) for their preparation was in a polyoxyethylene (20) sorbitan monooleate-to-sorbitan monooleate ratio of 3:2. “Blank” nanodispersions were developed as follows: the aqueous phase (double-distilled water and polyoxyethylene (20) sorbitan monooleate) was heated to 80 ± 2 °C and added dropwise to the lipid phase (a mixture of glyceryl behenate, almond or borage oil, and sorbitan monooleate), which was pre-heated to the same temperature, under continuous stirring at a speed of 750 rpm for 3 min. The resulting macroemulsion was homogenized for 3 min at 10,000 rpm and then ultrasonified for 15 min at 25 °C. Hyperforin-rich St. John’s wort extract-loaded samples were prepared by incorporating the extract (1.25% *w*/*w*) into the lipid phase before the emulsification under dark conditions.

#### 4.2.3. Microbial Strains

Reference strains of *Escherichia coli* ATCC 25922, *Staphylococcus aureus* ATCC 25923, *Pseudomonas aeruginosa* ATCC 10145, *Klebsiella pneumoniae* ATCC 10031, and *Candida albicans* ATCC 10231 were provided as MicroSwabs^®^ from Ridacom, Bulgaria. The microbial strains were activated according to manufacturer’s instructions, initially plating them on blood agar (HiMedia, supplied by Ridacom, Bulgaria). Prior to preparing the bacterial inoculum, each bacterial strain was cultured for 24 h in Brain Heart Infusion broth (HiMedia, supplied by Ridacom, Bulgaria) and subsequently plated again on agar medium. Single colonies were transferred to sodium chloride solution to obtain a suspension with turbidity corresponding to 0.5 McFarland standard (Grant Bio DEN-1, Grant Instruments, Cambridge, UK).

#### 4.2.4. Cells

MDBK cells were provided by the National Bank for Industrial Microorganisms and Cell Cultures in Sofia, Bulgaria. They were grown in DMEM medium (Gibco, Grand Island, NY, USA) supplemented with 10% FCS (Gibco BRL, Paisley, Scotland, UK), 10 mM HEPES buffer (AppliChem GmbH, Darmstadt, Germany), and antibiotics (100 IU/mL penicillin, 100 μg/mL streptomycin) (Sigma-Aldrich Chemie GmbH, Taufkirchen, Germany) at 37 °C and 5% CO_2_ atmosphere (Radobio Scientific Co., Ltd., Shanghai, China).

#### 4.2.5. Virus

Victoria strain of HSV-1 was obtained from the National Center for Infectious and Parasitic Diseases (Sofia, Bulgaria) and was propagated in a confluent monolayer of MDBK cells in DMEM (Gibco BRL, Paisley, Scotland, UK) with 0.5% fetal calf serum (Gibco BRL, Scotland, UK) and antibiotic mix added. After incubation at 37 °C and 5% CO_2_, viral stock was collected, aliquoted, and stored at −80 °C. The infectious titer was determined to be 10^8.5^ CCID_50_/mL.

#### 4.2.6. Reference Substance

ACV was kindly provided by the Deutsches Kresforschung Zentrum, Heidelberg, with a stock concentration of 3.0 mg/mL solution in DMSO, which was diluted in DMEM maintenance medium to the required concentration.

#### 4.2.7. Cytotoxicity Assay

Confluent MDBK cell monolayers grown in 96-well plates (Costar^®^, Corning Inc., Kennebunk, ME, USA) were treated with 0.1 mL/well of the test samples (“blank” and loaded with HP-rich extract NLCs), diluted 10-fold in DMEM maintenance medium, and then were placed at 37 °C and 5% CO_2_ for 48 h. After microscopic evaluation, the test sample was removed and its influence on the cell monolayer viability was determined by Neutral Red (Sigma-Aldrich Chemie GmbH, Germany) Uptake Assay, as described previously, and read spectrophotometrically [53]. The 50% cytotoxic concentration (CC_50_) was estimated for the HP-loaded specimens as the extract’s dose that reduced cell survival by 50% compared with untreated cells. Each sample was tested in triplicate, with four wells per test sample. Maximum tolerable concentration (MTC) of the hyperforin-rich HP extract in each NLC specimen was the highest concentration (lowest dilution) at which the cell monolayer was not affected and the treated cells’ appearance was similar to the cells in the non-treated control.

#### 4.2.8. Determination of Infectious Viral Titers

After MDBK cells reached confluence in 96-well plates, they were infected with 0.1 mL of virus suspension at 10-fold falling dilutions. Following 1 h, the unadsorbed virus was discarded and 0.1 mL/well DMEM medium was added to the cells. The plates were incubated at 37 °C and 5% CO_2_ in CO_2_ incubator (Radobio Scientific Co., Ltd., Shanghai, China) for 48 h. Infectious viral titer was determined by microscopic observation of the cell monolayer and cytopathic effect (CPE) determination. Visually determined CPE was confirmed by Neutral Red Uptake Assay as described in the previous section. Viral titers are expressed as lg IU (infectious units) (CCID_50_)/mL.

#### 4.2.9. Antiviral Activity Assay

Confluent MDBK cells in 96-well plates were inoculated with 100-cell culture infectious dose of 50% (CCID_50_) in 0.1 mL. After 60 min of viral adsorption to host cells, the unadsorbed virus was removed and the test sample was added following 10-fold dilution step. Cells were incubated for 48 h at 37 °C and 5% CO_2_. The CPE was determined by the Neutral Red Uptake Assay, and the percentage of CPE inhibition for each concentration of the test sample was calculated by an established calculation protocol [53]. The concentration at which 50% of the cytopathic effect was inhibited compared with the mock-control was defined as the inhibitory concentration 50% (IC_50_). From the obtained IC_50_ and toxicity CC_50_ values determined so far, the SI for the HP extract in each loaded NLC sample was determined using the CC_50_/IC_50_ ratio.

#### 4.2.10. Virucidal Activity (Effect on Extracellular Virions)

Samples containing virus (10^5^ CCID_50_) and test sample at its MTC were mixed in a 1:1 ratio and then stored at 20 °C for 15, 30, 60, 90, and 120 min intervals. The residual infectious virus content in each sample was then determined by titration [54], and the reduction in viral titer in samples containing the test products compared with the untreated viral control was assessed by determining the difference in viral titers ∆lg (delta log).

#### 4.2.11. Effect on Viral Adsorption

Monolayers of MDBK cell culture in 24-well plates were pre-cooled to 4 °C and inoculated simultaneously with 10^4^ CCID_50_ of HSV-1 and MTC of the test sample. This was followed by incubation at 4 °C for the time of virus adsorption. At various time intervals (15, 30, 45, and 60 min), cells were washed with PBS to remove both test components and unattached virus, and then covered with maintenance medium and incubated at 37 °C for 24 h. Triple freezing and thawing of the obtained samples was followed by an estimation of the infectious virus titer of each sample by an end-point dilution method. ∆lg was calculated on the basis of differences in viral titers compared with the viral control (untreated with the products). Each sample was prepared in quadruplicate.

#### 4.2.12. Antibacterial and Antifungal Activity

The antibacterial and antifungal activities of the NLC samples, “blank” and loaded with HP-rich extract, were tested against *E. coli*, *S. aureus*, *P. aeruginosa*,* K. pneumoniae*, and *C. albicans*. The starting extract concentration in the extract-loaded samples (HP-NLC1 and HP-NLC2) was 12.5 mg/mL. A serial 2-fold dilution of the samples was carried out in 1.0 mL Mueller–Hinton broth-containing sterile tubes up to 1:256 dilution of the stock solutions. The vials were then inoculated with 0.1 mL standardized to 0.5 McFarland turbidity equivalent microbial suspension corresponding to cell count density of approximately 1.5 × 10^8^ CFU/mL. Samples for positive control were set by combining 0.1 mL of each microbial suspension with 1.0 mL Mueller–Hinton broth; non-inoculated mixtures of the test samples with Mueller–Hinton broth in a 1:1 ratio were left as negative controls. The antimicrobial susceptibility of all strains was assessed by using chlorhexidine diacetate as a control inhibitor. All experiments were repeated in triplicate. The test vials contaminated with bacterial strains were incubated for 24 h at 37 °C, and the ones contaminated with *C. albicans* for 48 h at 35 °C. The MBC and MFC were determined by transferring all test suspensions in a single-bacterial-loop-volume onto blood agar. The so-obtained specimens were incubated once again under the same conditions as described above. The lowest concentration at which bacterial/fungal growth appeared to be inhibited at ≥99.9% was reported as MBC/MFC.

#### 4.2.13. Statistical Analysis

The values of CC_50_ were calculated using non-linear regression analysis (GraphPad 4 Software, San Diego, CA, USA). The values were presented as means ± SD from three independent experiments. The difference significance between the cytotoxicity values of the samples and the reference substance ACV, as well as between the effects of the test products on the viral replication, was performed by Student’s *t*-test, where *p*-values of < 0.05 were considered significant.

## 5. Conclusions

In our study, we investigated the anti-herpesvirus and antimicrobial properties of NLC-incorporated St. John’s wort extract rich in hyperforin. We reported slight activity against HSV-1, which covered viral replication but did not affect either extracellular virions or viral adsorption. By comparing the effects of the loaded and the “blank” NLCs, we also proved the latter is owing to the extract. The antimicrobial properties of the HP-NLCs were also assessed as being present due to the extract itself. A more pronounced activity was exhibited by HP-NLC2. The latter was highly effective against *K. pneumoniae* and *C. albicans*, while its activity against *S. aureus* was manifested at much higher extract concentrations. Assuming all obtained results, we can conclude that HP-NLC2 serves as a more efficient nanocarrier of the hyperforin-rich St. John’s wort extract.

## Figures and Tables

**Figure 1 pharmaceuticals-18-00366-f001:**
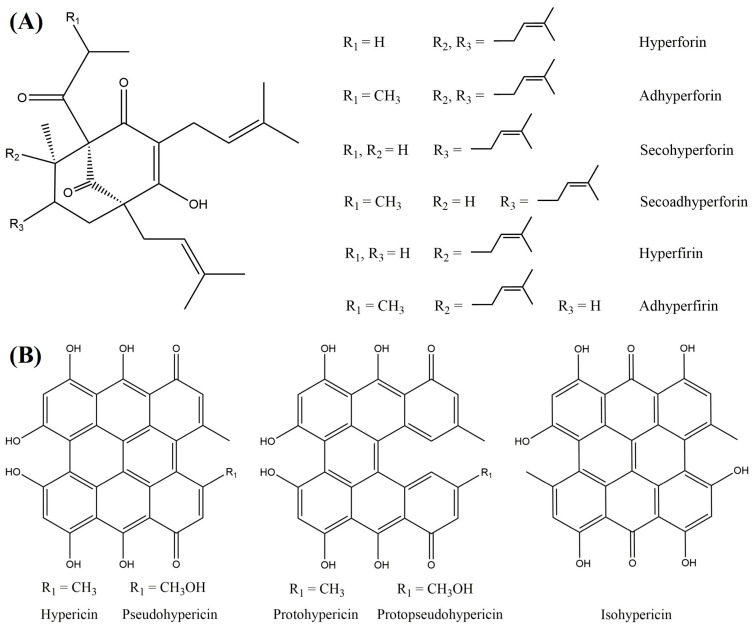
Chemical structures of the main phloroglucinols (**A**) and naphthodianthrones (**B**) present in *Hypericum perforatum* L.

**Figure 2 pharmaceuticals-18-00366-f002:**
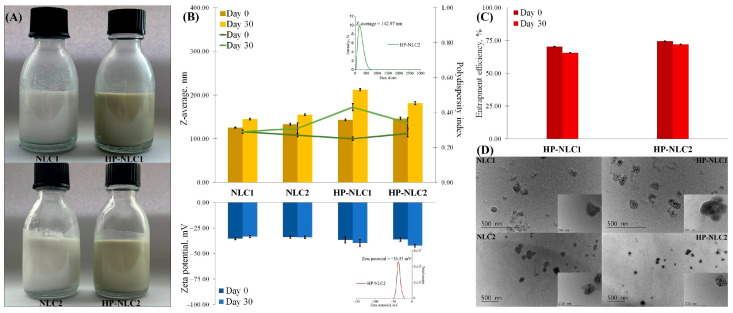
Physicochemical characteristics of “blank” and hyperforin-rich HP extract-loaded lipid nanoparticles. Subfigure (**A**) presents the visual appearance of the obtained lipid nanodispersions. The extract-loaded ones were stored in amber glass vials but were photographed in transparent ones to display their qualitative characteristics. Mean particle size (Z-average), polydispersity index, zeta potential (**B**), and entrapment efficiency (**C**) of the nanoparticles were assessed immediately after their preparation and after 30-day storage at 4 °C. Subfigure (**D**) demonstrates the shape and inner morphology of the nanocarriers. Data from the study of Sotirova et al. [32] were used for constructing the figure.

**Figure 3 pharmaceuticals-18-00366-f003:**
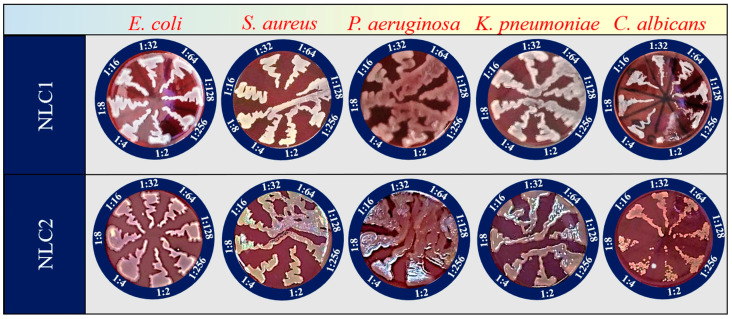
Antibacterial and antifungal activity of the “blank” NLC samples.

**Figure 4 pharmaceuticals-18-00366-f004:**
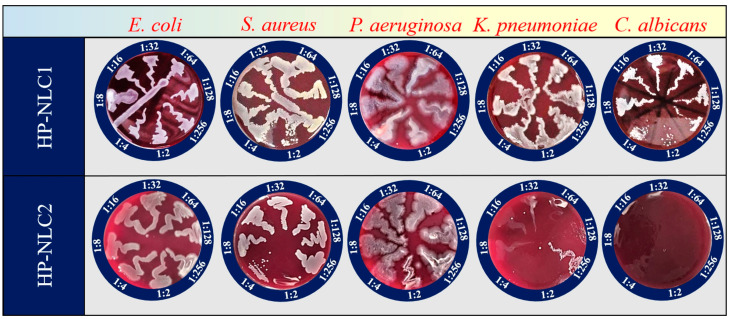
Antibacterial and antifungal activity of the HP-NLC samples.

**Table 1 pharmaceuticals-18-00366-t001:** Cytotoxicity and antiviral activity of the HP-loaded test samples.

Sample	MDBK Cell Line		HSV-1 (Victoria Strain)	
	CC_50_ Mean ± SD (μg/mL)	MTC (μg/mL)	IC_50_ Mean ± SD (μg/mL)	SI
HP-NLC1	87.5 ± 2.3 **	12.5	25.3 ± 2.2 **	3.5
HP-NLC2	94.7± 3.6 *	12.5	23.7 ± 2.1 **	4.0
ACV	291.0 ± 9.4	nd	0.33 ± 0.03	881.8

CC_50_—cytotoxic concentrations 50%; SD—standard deviation; MTC—maximum tolerable concentration; IC_50_—inhibitory concentration 50%; SI—selectivity index, which was calculated from the CC_50_/IC_50_ ratio; nd—no data; * *p* < 0.05—when comparing the value of each compound with the reference substance ACV; ** *p* < 0.001—when comparing the value of each compound with the reference substance ACV.

**Table 2 pharmaceuticals-18-00366-t002:** Virucidal activity of the test samples against HSV-1 virions (Victoria strain).

Sample	Δlg				
	15 min	30 min	45 min	60 min	120 min
NLC1	0.5	0.5	0.5	0.5	0.5
HP-NLC1	1.0	1.0	1.0	1.0	1.0
NLC2	0.5	0.5	0.5	0.5	0.5
HP-NLC2	0.5	0.5	0.75	0.75	0.75
70% ethanol	8.0	8.0	8.0	8.0	7.75

**Table 3 pharmaceuticals-18-00366-t003:** Influence of the test samples on the stage of adsorption of HSV-1 (Victoria strain) to sensitive MDBK cells.

Sample	Δlg			
	15 min	30 min	45 min	60 min
NLC1	0.25	0.25	0.25	0.5
HP-NLC1	0.25	0.25	0.5	0.5
NLC2	0.0	0.25	0.25	0.5
HP-NLC2	0.25	0.25	0.5	0.75

## Data Availability

The data presented in this study are available on request from the corresponding author.

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
