# Peer review of "Antimicrobial and Antiherpetic Properties of Nanoencapsulated *Hypericum perforatum* Extract"

_pharmaceuticals, 2025, doi:10.3390/ph18030366_

Round 1
Reviewer 1 Report
Comments and Suggestions for Authors
Review Report
Antimicrobial and Antiherpetic Properties of Nanoencapsulated Hypericum perforatum L. Extract
Comments for Authors: Sotirova et al.
- Title: Replace "St. John’s Wort" with Hypericum perforatum L.
- Novelty of the Study: Clearly state the novelty of this research in comparison to previous studies on nanoencapsulated Hypericum perforatum L.
- Microbial Strain Identification:
Line 19: Mention the ATCC numbers for Staphylococcus aureus, Escherichia coli, Pseudomonas aeruginosa, Klebsiella pneumoniae, and Candida albicans used in the study.
- Abstract: At the end, briefly add the utility and future prospects of the study.
- Keywords: Avoid listing microbial names in the keywords section.
- Literature References:
Lines 42-44: "The technological evolution in scientific research satisfies the growing interest in the herb; multiple biochemical, pharmacological, pharmacokinetic, and clinical trials have been conducted and identified its antioxidant, anti-inflammatory, antibacterial, and antiviral properties, as well as the mechanisms..."
Cite relevant references for each property mentioned.
- Microbial Strain Information:
Lines 53-56: Provide the strain numbers for each microorganism used.
- Inoculum Preparation:
Lines 143-145: Briefly explain the revival, inoculum preparation, and density determination using the McFarland standard.
- Terminology Correction:
Line 228: Replace "contaminated" with "inoculated."
- Growth Conditions:
Explain why Candida albicans was grown at 35°C when its optimal growth condition is generally 25°C.
- Minimum Bactericidal/Fungicidal Concentration:
Lines 232-234: "The lowest concentration at which bacterial/fungal growth appeared to be inhibited at 99.9% was reported as MBC/MFC."
How was 99.9% inhibition ensured? Was a plate assay conducted? Please clarify.
- Figures
Figure 1: Include standard error bars.
Figures 2 and 3: Use original images instead of edited versions.
- Experimental Design:
The study includes bacterial strains and only one fungal strain (Candida albicans). To support claims of antifungal activity, we suggest testing at least two or three additional fungal strains.
Justify why blood agar was used for the final assessment.
- Control Experiments:
Specify the known inhibitor control used in antimicrobial testing.
- Graphical Abstract:
Line 436: The acknowledgment states that a graphical abstract was created in BioRender. However, no graphical abstract is present in the manuscript. Please include it if applicable.
Comments on the Quality of English Language
The English could be improved to more clearly express the research.
Author Response
Dear reviewer,
Thank you very much for your thorough work on our manuscript and for your contribution! In the current document, a point-by-point response to your comments and suggestions is provided. You can find the changes in the manuscript regarding your recommendations highlighted in yellow.
Q1. Title: Replace "St. John’s Wort" with Hypericum perforatum L.
R1. Done.
Q2. Novelty of the Study: Clearly state the novelty of this research in comparison to previous studies on nanoencapsulated Hypericum perforatum L.
R2. A final conclusive sentence was added in the Abstract concerning your comment.
Q3. Microbial Strain Identification: Line 19: Mention the ATCC numbers for Staphylococcus aureus, Escherichia coli, Pseudomonas aeruginosa, Klebsiella pneumoniae, and Candida albicans used in the study.
R3. The ATCC numbers of the strains used in this study are specified in Materials and Methods/ Microbial strains. Following your request, we added the numbers in the abstract as well.
Q4. Abstract: At the end, briefly add the utility and future prospects of the study.
R4. A final conclusive sentence was added in the Abstract concerning your comment.
Q5. Keywords: Avoid listing microbial names in the keywords section.
R5. The microbial names were removed from the keywords list.
Q6. Literature References: Lines 42-44: "The technological evolution in scientific research satisfies the growing interest in the herb; multiple biochemical, pharmacological, pharmacokinetic, and clinical trials have been conducted and identified its antioxidant, anti-inflammatory, antibacterial, and antiviral properties, as well as the mechanisms...". Cite relevant references for each property mentioned.
R6. Proper references were added.
Q7. Microbial Strain Information: Lines 53-56: Provide the strain numbers for each microorganism used.
R7. The ATCC numbers of the strains used are specified in Materials and Methods/ Microbial strains. Lines 53-56, particularly, regard summarized information from other researchers’ reports.
Q8. Inoculum Preparation: Lines 143-145: Briefly explain the revival, inoculum preparation, and density determination using the McFarland standard.
R8. The microbial strains were activated following the manufacturer's instructions, initially plating them on Blood agar (HiMedia, supplied by Ridacom, Bulgaria). Prior to preparing the bacterial inoculum for the tests in this study, each bacterial strain was cultured for 24 hours in Brain Heart Infusion broth (HiMedia, supplied by Ridacom, Bulgaria) and subsequently plated again on agar medium. This procedure aimed to ensure the maximum viability of the strains. For the preparation of the bacterial inoculum, as described in the Materials and Methods section, single colonies were transferred to sodium chloride solution to obtain a suspension with turbidity corresponding to 0.5 McFarland standard (Grant Bio DEN-1, UK), which approximately corresponds to cell count density of 1.5 × 10^8 cells.
Q9. Terminology Correction: Line 228: Replace "contaminated" with "inoculated."
R9. Done.
Q10. Growth Conditions: Explain why Candida albicans was grown at 35°C when its optimal growth condition is generally 25°C.
R10. The incubation of Candida albicans was performed following the guidelines outlined in the Clinical Laboratory Standards Institute (CLSI): Reference Method for Broth Dilution Antifungal Susceptibility Testing of Yeast; Approved Standard—Third Edition. CLSI document M27-A3; 2008.
For antimicrobial activity testing in broth media, the prescribed guidelines include the following procedures: "Incubate the microdilution plates at 35 °C and observe for the presence or absence of visible growth. …The original work includes reading the MIC after 48 hours of incubation"
Regarding the cultivation of Candida albicans on agar media, the HiMedia instruction sheet specifies the following incubation conditions: Candida albicans ATCC 10231 - Incubation temperature: 30–35°C - Incubation period: 24–48 hours. A link to the referenced guideline is provided below: [Available online: https://www.himedialabs.com/as/mph063-sabouraud-dextrose-agar-plate.html]
Q11. Minimum Bactericidal/Fungicidal Concentration: Lines 232-234: "The lowest concentration at which bacterial/fungal growth appeared to be inhibited at 99.9% was reported as MBC/MFC."How was 99.9% inhibition ensured? Was a plate assay conducted? Please clarify.
R11. Thank you for your comment. We believe that replacing "99.9%" with "≥99.9%" is more appropriate. In our case, microbial growth was completely inhibited, with no visible colonies on the agar media. For this assessment, we used blood agar. The described method is applied following the guidelines outlined in the Clinical Laboratory Standards Institute (CLSI): Methods for determining bactericidal activity of antimicrobial agents, Approved guideline. CLSI document M26-A [Available on-line: https://clsi.org/media/1462/m26a_sample.pdf]
Citation: “The minimal concentration of drug needed to kill most (≥99.9%) of the viable organisms after incubation for a fixed length of time under a given set of conditions of bactericidal activity is known as the minimal bactericidal concentration (MBC)”.
Q12. Figures. Figure 1: Include standard error bars.
R12. The error bars in the figure were bolded.
Q13. Figures 2 and 3: Use original images instead of edited versions.
R13. The figures contain original images. The template used (“clock” style blue rings) is also original as it was created by one of the authors. We believe this template allows a better understanding of the dilutions and the active concentrations.
Q14. Experimental Design: The study includes bacterial strains and only one fungal strain (Candida albicans). To support claims of antifungal activity, we suggest testing at least two or three additional fungal strains.
R14. Thank you for this remark, we will consider performing a following study and include a wider fungal spectrum to support the obtained data. As for the present study, we are not able to perform additional experiments in the short term since other fungal strains are currently not available in our laboratory.
Q15. Justify why blood agar was used for the final assessment.
R15. Blood agar is used for determining the Minimum Bactericidal Concentration (MBC) for several reasons related to its specific characteristics and advantages in microbiological testing:
– Supports the growth of a wide range of microorganisms: Blood agar contains hemoglobin and other nutrients that provide optimal conditions for the growth of a broad spectrum of bacteria, including pathogenic microorganisms. This makes it suitable for tests that require full expression of bacterial growth.
– Better assessment of bacterial viability
– Supports strain stability: Blood agar provides a stable environment, minimizing distortion of results that may occur when using poorer media that lack sufficient nutrients to sustain bacterial growth over a longer period.
Q16. Control Experiments: Specify the known inhibitor control used in antimicrobial testing.
R16. The microbial strains used in this study were previously treated with chlorhexidine as a control compound. Following your remark, we included the data in the main text.
Q17. Graphical Abstract: Line 436: The acknowledgment states that a graphical abstract was created in BioRender. However, no graphical abstract is present in the manuscript. Please include it if applicable.
R17. The graphical abstract was attached later by request of the editors. It is possible that on the day of your revision, the GA was still not available. I hope you will have the accessibility to it in the second round of revision.
In addition, I would like to assure you that the revised manuscript was edited by a professional linguist and a signed statement was attached.
Reviewer 2 Report
Comments and Suggestions for Authors
Antimicrobial and Antiherpetic Properties of Nanoencapsulated St. John’s Wort Extract
- Keywords are essential for discoverability. I recommend reviewing and refining them to ensure they align with standard practices in published literature. Looking at keywords in similar published papers might be helpful.
“Keywords: antibacterial activity; antiviral effects; Candida albicans; herpes virus; Hypericum
perforatum; hyperforin; Klebsiella pneumoniae; lipid nanoparticles; nanostructured lipid carriers; Staphylococcus aureus”
- Add suitable references to the introduction section, lines 42-45
- On page 2, line 76, the statement is made that the nanoparticles are 'more stable than liposomes.' Could the authors please clarify which aspect of stability is being referred to here (e.g., storage stability, stability in biological fluids, etc.)?
- Adding/drawing the chemical structures of representative phloroglucinols and naphthalenediones to the Introduction section would enhance the manuscript and make it more appealing to readers
- Define 'Δlg' at its first use. Consider using either the full phrase 'delta log' or the abbreviation with the definition in parentheses (e.g., Δlg (delta log)).
- The manuscript mentions several advanced techniques (DLS, laser Doppler electrophoresis, EE determination, ATR-FTIR spectroscopy, XRD analysis, and TEM) to characterize the HP-loaded NLCs. However, the figures and detailed explanations of these results are not present in the main text. Including this data, perhaps as supplementary figures and tables, would significantly strengthen the manuscript and allow for a more thorough evaluation of the NLC characterization.
Preparation of “blank” and HP-loaded NLCs
The optimal carriers for the thus-obtained extract were identified based on a
thorough physicochemical characterization of twenty potential specimens by Dynamic
Light Scattering (DLS), Laser Doppler Electrophoresis, Entrapment Efficiency (EE)
determination, Attenuated Total Reflectance (ATR)-FTIR Spectroscopy, X-ray Diffraction
analysis, and Transmission Electron Microscopy (TEM).
Figure 1's caption mentions DLS and zeta potential. To support the discussion of stability and size, provide representative DLS and zeta potential images/figures. This would allow readers to directly assess the stability and size characteristics of the NLCs
Author Response
Dear reviewer,
Thank you very much for your thorough work on our manuscript and for your contribution! In the current document, a point-by-point response to your comments and suggestions is provided. You can find the changes in the manuscript regarding your recommendations highlighted in green.
Q1. Keywords are essential for discoverability. I recommend reviewing and refining them to ensure they align with standard practices in published literature. Looking at keywords in similar published papers might be helpful.
R1. Thank you for this remark. The keywords were revised.
Q2. Add suitable references to the introduction section, lines 42-45
R2. Done. Proper references were added.
Q3. On page 2, line 76, the statement is made that the nanoparticles are 'more stable than liposomes.' Could the authors please clarify which aspect of stability is being referred to here (e.g., storage stability, stability in biological fluids, etc.)?
R3. Done.
Q4. Adding/drawing the chemical structures of representative phloroglucinols and naphthalenediones to the Introduction section would enhance the manuscript and make it more appealing to readers
R4. Done.
Q5. Define 'Δlg' at its first use. Consider using either the full phrase 'delta log' or the abbreviation with the definition in parentheses (e.g., Δlg (delta log)).
R5. Done.
Q6. The manuscript mentions several advanced techniques (DLS, laser Doppler electrophoresis, EE determination, ATR-FTIR spectroscopy, XRD analysis, and TEM) to characterize the HP-loaded NLCs. However, the figures and detailed explanations of these results are not present in the main text. Including this data, perhaps as supplementary figures and tables, would significantly strengthen the manuscript and allow for a more thorough evaluation of the NLC characterization.
R6. We aimed to summarize the most important results from these studies as they are not an object to the current research and are already published elsewhere. Citations to these publications of ours were added so that the readers can find the complete data regarding the results from these tests by tracking the references in question. Following your recommendation, we extended the information in Figure 1.
Q7. Figure 1's caption mentions DLS and zeta potential. To support the discussion of stability and size, provide representative DLS and zeta potential images/figures. This would allow readers to directly assess the stability and size characteristics of the NLCs
R7. We extended Figure 1 with respect to your comment.
Round 2
Reviewer 1 Report
Comments and Suggestions for Authors
No further comments
Author Response
Thank you!
Reviewer 2 Report
Comments and Suggestions for Authors
The revised manuscript is now considered acceptable.
Author Response
Thank you!